# Preparation and Properties of Municipal Solid Waste Incineration Alkali-Activated Lightweight Materials through Spontaneous Bubbles

**DOI:** 10.3390/polym14112222

**Published:** 2022-05-30

**Authors:** Yongyu Li, Hongxue Zhang, Guodong Huang, Yi Cui, Jiacheng Feng, Yuting Zhang, Dawei Li, Jielei Zhu

**Affiliations:** 1School of Civil Engineering and Construction, Anhui University of Science and Technology, Huainan 232001, China; yyli@aust.edu.cn (Y.L.); gdhuang@aust.edu.cn (G.H.); ycui@aust.edu.cn (Y.C.); jcfeng@aust.edu.cn (J.F.); ytzhang@aust.edu.cn (Y.Z.); dwli@aust.edu.cn (D.L.); jlzhu@aust.edu.cn (J.Z.); 2School of Mechanics and Optoelectronic Physics, Anhui University of Science and Technology, Huainan 232001, China; 3Hefei Comprehensive National Science Center, Institute of Energy, Hefei 230031, China; 4Institute of Environment-Friendly Materials and Occupational Health, Anhui University of Science and Technology, Wuhu 241003, China; 5Engineering Quality Inspection and Safety Evaluation, Fujian University Engineering Research Center, Longyan 364000, China

**Keywords:** municipal solid waste incineration, alkali-activated, lightweight materials, self-foaming, compressive strength

## Abstract

A self-foaming alkali-activated lightweight material was prepared by the pretreatment of municipal solid waste incineration bottom ash (BA). The low weight could be achieved without adding a foaming agent by using the low-density and self-foaming expansion characteristics of BA in combination with a strong alkali. The effects of BA, liquid sodium silicate (LSS), and calcium hydroxide (CH) on dry and wet densities, as well as water absorption, are discussed. The results show that increasing the BA content can significantly improve the foaming effect and reduce the dry and wet densities of specimens. However, it also leads to a sudden decrease in compressive strength and a significant increase in water absorption. LSS and CH can significantly improve the ability to seal bubbles by accelerating condensation, and they further reduce dry and wet densities without significantly improving water absorption. It is most effective at BA, LSS, and CH contents of 60, 20, and 2%, respectively.

## 1. Introduction

With the continuous development and advancement of building technology and material science, structural systems are developing toward super high-rise and long-span buildings. This trend imposes higher requirements on the strength and density of non-load-bearing structures and necessitates the development of partition materials with lightweight, high strength, green energy conservation, and environmental protection characteristics [1]. Traditional lightweight materials usually employ Portland cement as their main binding material body, where an appropriate amount of foaming agent is added to achieve the lightweight property. Furthermore, large amounts of water-reducing agents and accelerators are added to meet strength and setting time requirements. However, the production of cement is associated with serious problems of high energy consumption and high pollution [2]. At the same time, the use of foaming agents, superplasticizers, and accelerators significantly increase the preparation cost. Therefore, cement-based lightweight materials offer no advantages in terms of preparation cost, technical difficulty, material performance, low-carbon properties, energy conservation, and environmental protection [3]. With the proposal of the strategic goal of “carbon peak and carbon neutrality” and the attention focusing on energy conservation and environmental protection, many scientists are urgently striving to develop a lightweight material with these advantages to replace the demand for traditional lightweight materials [4].

In recent years, scholars have increasingly committed to the research and development of energy-saving and environmentally friendly lightweight materials [5]. A polymeric material is a type of inorganic non-metallic material with a three-dimensional network structure and is composed of AlO_4_ and SiO_4_ tetrahedral structural units [6]. This material exhibits not only excellent mechanical properties and acid–base resistance, fire resistance, and high-temperature resistance; but also significant advantages such as being low carbon, energy saving, and offering environmental protection by using industrial solid waste as the main raw material [7]. For example, Taherlou et al. [8] studied the feasibility of simultaneously using different percentages of municipal solid waste incineration (MSWI) bottom ash (BA) and the treated industrial wastewater in self-compacting concrete. Ferone et al. [9] studied three samples of municipal solid waste incineration fly ash (MSWI-FA) and stabilized them in systems containing coal fly ash to fabricate geopolymers through a polycondensation reaction. Czop and Ła’zniewska-Piekarczyk [10] studied a preliminary evaluation of the suitability of slag from the MSWI plant for its use as a replacement for cement.

Moreover, Ren et al. prepared magnesium phosphate cementitious material using potassium dihydrogen phosphate, and a new lightweight material was obtained by adding phosphogypsum and an air-entraining agent [11]. Using construction waste as the main material, a new lightweight material was prepared by adding aqueous polyacrylic acid resin and H_2_O_2_ solution [12]. A magnesium-based lightweight material was prepared by using a magnesium-based cementitious material as the main material and adding manganese dioxide, H_2_O_2_ solution, calcium stearate, and polyacrylamide [13]. However, although the obtained materials are not reliant on the consumption of Portland cement and the preparation of green and environmentally friendly lightweight materials has already been realized, it is still necessary to use a special foaming agent to lower the weight. This leads to an increase in both the cost and complexity of the preparation process of these lightweight materials, which cannot guarantee the widespread application of lightweight materials [14].

In this study, MSWI bottom ash (BA) was used as the main raw material and was supplemented with blast-furnace granulated slag, liquid sodium silicate (LSS), and hydrated lime powder as mineral additives. The lightweight, self-foaming, and expansion characteristics of BA in the case of alkali were fully utilized through sodium hydroxide excitation [15]. At the same time, the rapid setting characteristics of LSS were adopted to effectively isolate and seal bubbles, which thus provides an energy-saving, environmentally friendly, and non-foaming lightweight material [16]. If these materials are used as the partition wall material for non-load-bearing components in the building structure, the large-scale safe consumption and resource recycling of BA can be realized. Moreover, these materials can improve the preparation of high-performance, low-cost, low-carbon, green, energy-saving, and environmentally friendly BA lightweight materials, which can thus realize the replacement of cement lightweight materials.

## 2. Experimental Materials and Methods

### 2.1. Experimental Materials

#### 2.1.1. Municipal Solid Waste Incineration Bottom Ash Pretreatment

BA (from the Hefei domestic waste incineration power plant, Hefei, Anhui, China) was used as a binder to prepare the alkali-activated materials. BA was first processed into particles with a size below 0.1 mm using a jaw crusher and then milled with a ball mill (Qm-5 planetary ball mill provided by Beijing Haifuda Technology Co., Ltd., Beijing, China) for 30 min to obtain a micro powder with a particle size of less than 50 μm. The specific surface area of BA was greater than 410 m^2^ kg^−1^, and the median particle size was 37 μm. X-ray fluorescence (XRF) spectroscopy analysis (Table 1) showed that the silicon oxide content of the BA was 53.8%, and the contents of calcium and aluminum were slightly below 15%. Its micromorphology is shown in Figure 1a.

#### 2.1.2. Blast Furnace Granulated Slag

Blast furnace granulated slag (GBFS) (Xiangtai mineral products Co., Ltd., Shijiazhuang, China) was used as a binder to prepare the alkali-activated materials. The performance reached grade S95, which met the national standard for GBFS powder used in cement, mortar, and concrete (GB/T 18046-2017) [17]. The specific surface area of the GBFS exceeded 412 m^2^ kg^−1^, and the median particle size was 42 μm. The chemical composition is presented in Table 1, and its micromorphology is shown in Figure 1b.

#### 2.1.3. Calcium Hydroxide and Sodium Hydroxide

Both calcium hydroxide (CH) and sodium hydroxide (NH) used in this study were of analytical purity, with a purity exceeding 99.8%.

#### 2.1.4. Liquid Sodium Silicate

The LSS (Na_2_SiO_3_) used in this study was of industrial grade with purity exceeding 95%; therein, Na_2_O, SiO_2_, and H_2_O contents accounted for 9.68, 25.26, and 65.02% of the total mass, respectively.

#### 2.1.5. Other Ingredients

Grade 42.5 ordinary Portland cement was used, and the specific surface area of the cement particles reached 342 m^2^ kg^−1^. The density was 3.09 g cm^−3^, and the loss on ignition was less than 5%. Tap water was used as the experimental water. Sodium alcohol ether sulphate (RO(CH_2_CH_2_O)nSO_3_Na, Shandong Linyi LUSHEN Chemical Co., Ltd., Linyi, China) was adopted as a liquid foaming agent, with a foaming factor of 30 times and a settlement of 6 mm. The content of active substance was more than 72%, the content of free oil was less than 3.5%, and the content of sodium sulfate was less than 1.5%.

### 2.2. Preparation and Curing of Test Specimens

#### 2.2.1. Experimental Mix Proportion

Table 2 presents the experimental mix proportions. LYY-1 is the cement control group without a foaming agent. Groups LYY-2 to LYY-4 were mixed with a foaming agent at different proportions to test the influence of the foaming agent on various properties. In groups LYY-5 to LYY-7, the contents of BA were increased gradually, and the foaming effect of BA was tested. The sealing effect of LYY-8 and LYY-10 on bubbles was gradually improved. Groups LYY-11 to LYY-13 were mixed with different proportions of CH to test the sealing effect of CH on bubbles (the liquid–solid ratio was 0.5).

#### 2.2.2. Specimen Preparation and Curing

For the preparation and curing method of the samples, refer to the Chinese standard method for testing cement mortar strength (ISO method, GB/T 17671-2021) [18]. Considering LYY-13 as an example, first, the NH was dissolved in water according to the proportion shown in Table 2. Then, BA, GBFS, and CH were mixed evenly. The NH solution was added, and the mixture was slowly stirred for 15 s, followed by vigorous stirring for 15 s. Next, the LSS was added, and the contents were vigorously stirred for 30 s, and poured into a 40 mm × 40 mm × 160 mm mold. The mold was left to set in a curing room and removed after 3 days (to reduce the impact of mold removal on the specimen integrity). After the removal of the mold, the specimen was kept in the curing room until the aging of the experimental design at a curing temperature of 20 ± 2 °C was completed. The humidity exceeded 95%.

### 2.3. Experimental Method

#### 2.3.1. Compressive Strength Test

A fully-automatic compression and bending machine (DYE-300S, Beijing Zhongtong Jianyi Instrument Equipment Co., Ltd., Beijing, China) was used to test the compressive strength of the specimens after curing for 3, 28, and 60 days.

#### 2.3.2. Determination of Dry and Wet Density and Water Absorption

First, the volume of the specimen was measured: the three axis dimensions of length, width, and height were measured to calculate the volume at an accuracy of 1 mm (the size of the specimen was 40 mm × 40 mm × 160 mm). Second, the wet density was measured: the mass of the specimen in its natural state was weighed to calculate the wet density at an accuracy of 0.1 g (according to the following formula: wet density = mass/volume in the natural state). Then, the dry density was measured, for which the specimen was placed in an electric blast drying oven, dried continuously at 105 ± 5 °C for 24 h, and weighed (accuracy of 0.1 g). The dry density was calculated (according to the formula: dry density = mass/volume after drying). Finally, the water absorption was measured, for which the specimen was put into a water tank at 20 ± 5 °C, and water was added to 1/3 of the height of the specimen. The specimen was soaked for 24 h; then, water was added to 2/3 of the height of the specimen, followed by soaking for 24 h. Finally, water was added to 30 mm above the specimen, followed by soaking for another 24 h. The specimen was then wiped dry and weighed immediately. The water absorption was calculated (according to the following formula: water absorption = water-saturated mass − mass in a dry state/ass in dry state). The average value of the three specimens was taken as the result of the abovementioned experiments.

#### 2.3.3. Scanning Electron Microscopy

An OST-AF200 HD video microscope (Suzhou oester Optical Instrument Co., Ltd., Suzhou, China) and scanning electron microscopy (SEM; FEI QuantaTM 250, Washington, DC, USA) were used to observe the pore structure distribution, bubble distribution, density, and crack propagation of the specimens (after 28 days). The SEM analysis was performed in a low vacuum mode under low nitrogen pressure.

## 3. Results and Discussion

### 3.1. Analysis of Foaming and Expansion Mechanism of Bottom Ash in the Case of Alkali

Figure 2a demonstrates that after crushing and ball milling, BA becomes mainly grayish black. When BA was added to the NH solution (see LYY-5 in Table 2 for the dosage of BA, NH, and water) and mixed to form a uniform black slurry (Figure 2b), bubbles immediately emerged from the slurry. This resulted in the significant expansion of the slurry. Through collecting the gas in the bubbles and assessing it with a meteorological mass spectrometer, the gas produced was identified as mainly consisting of hydrogen (more than 95%). This is attributed to the fact that domestic waste originates from a wide range of sources and contains a large number of aluminum products. Elemental aluminum oxidizes in the air and forms a dense alumina protective film [19]. As the melting point of alumina is above 2000 °C, it ensures that the elemental aluminum wrapped in alumina is not oxidized during the calcination of municipal solid waste (calcination temperature 800 °C) [20]. When BA is added to the NH solution, the alumina protective film in BA dissolves rapidly, and the internal elemental aluminum reacts with the NH solution, thus generating a large amount of hydrogen. Eventually, this leads to foaming and the expansion of BA [21].

The appearance and internal structure of the lightweight alkali-excitation specimen prepared from BA are shown in Figure 2c,d. Pores of different shapes and sizes are densely distributed across the surface and continue to extend to the interior of the specimen. The entire specimen is characterized by a fluffy sponginess, which significantly reduces the density. Moreover, the interior of the specimen is also not dense. It contains a large number of bubbles and pores, which are caused by the foaming and expansion of BA in the presence of alkali. Therefore, by making full use of the foaming and expansion characteristics of BA in the presence of alkali, lightweight specimens can be prepared from BA without the need to add foaming agents. Furthermore, the preparation cost can be significantly reduced, and the preparation process can be simplified, which results in a high-quality method for the large-scale resource utilization of BA.

### 3.2. Analysis of Compressive Strength

#### 3.2.1. Analysis of Compressive Strength of Cement Specimens

The development law for the compressive strength of cement specimens is shown in Figure 3a. The compressive strength of specimen LYY-1 cured for 3, 28, and 60 days reached 16.5, 48.6, and 53.3 Mpa, respectively. When no foaming agent was added, specimen LYY-1 exhibited good compressive strength. However, when 10 g (1%) foaming agent was added, the compressive strength of specimen LYY-2 after curing for 3 days reached only 2.4 Mpa. Even if the specimen was cured for 28 and 60 days, the compressive strength increased only to 5.8 and 6.2 MPa, respectively. The addition of a foaming agent led to a sudden decrease in the compressive strength. The foaming agent led to the formation of a large number of bubbles in the specimen, which reduced the overall density; however, the brittle and porous structure also led to a sudden drop in compressive strength.

With an increased amount of foaming agent, the compressive strengths of specimens LYY-3 and LYY-4 continued to decrease significantly. When the content of the foaming agent reached 30 g (3%), the compressive strength of LYY-4 cured for 28 and 60 days further decreased to 0.8 and 1.0 MPa, respectively. Although the addition of a foaming agent could introduce many bubbles and significantly reduce the density of the specimen, the collection of bubbles also caused severe defects. These cause the observed sudden drop in compressive strength, which seriously hinders the application of cement lightweight specimens in engineering.

#### 3.2.2. Analysis of Compressive Strength of BA Alkali-Activated Specimens

The compressive strength of BA-alkali activated specimens is shown in Figure 3b. When the content of BA was 40%, the compressive strength of specimen LYY-5 cured for 3, 28, and 60 days reached 6.4, 11.6, and 13.1 MPa, respectively, which are 166.7, 100, and 113.1% higher than those of specimen LYY-2 (10 g (1%) foaming agent). BA could foam and expand without a significant reduction in the compressive strength. The homogeneous foaming of BA could ensure the uniform distribution of bubbles and reduce the damage to the compressive strength. Therefore, the mechanical properties of the lightweight BA specimens are clearly better than those of the lightweight cement specimens.

With increasing BA content, the compressive strength of BA alkali-activated specimens decreased slowly; however, the decrease range was significantly lower than that of cement lightweight specimens. When the content of BA was increased to 60%, the compressive strength of LYY-7 cured for 3, 28, and 60 days reached 4.9, 9.1, and 10.2 MPa, respectively, which are 23.4, 21.6, and 22.1% lower than those of specimen LYY-5 (40% BA). Moreover, the amount of foaming agent was significantly increased with the increasing BA content, resulting in the increased expansion and the reduction in compressive strength while playing a positive role in reducing the specimens’ density.

#### 3.2.3. Effect of Liquid Sodium Silicate on Compressive Strength

With the addition of LSS, the compressive strength of specimens LYY-8 and LYY-9 decreased continuously (Figure 3c). When the content of LSS was increased to 200 g, the compressive strength of specimen LYY-9 decreased to 3.2 MPa (3 days), 6.0 MPa (28 days), and 6.8 MPa (60 days), which are 34.7, 34.1, and 33.3% lower than those of specimen LYY-7 (not mixed with LSS), respectively. Although the addition of LSS further reduced the compressive strength, LSS could accelerate the condensation of the specimen [22] and significantly improve the bubble-sealing ability, which is beneficial for further reducing the density of the specimens. However, when the content of LSS was increased to 300 g, the compressive strength of specimen LYY-10 did not decrease significantly at any curing age (compared with specimen LYY-9). This shows that increasing the LSS content cannot continually improve the ability of bubble sealing, and a content of 200 g works the best.

#### 3.2.4. Effect of Calcium Hydroxide on Compressive Strength

At a CH content of 20 g (2%), the compressive strength of specimen LYY-11 increased to 3.7, 6.6, and 7.6 MPa, and these values are 15.6, 10.0, and 11.8% higher than those of specimen LYY-9 (without CH, Figure 3d). The addition of CH can significantly improve the compressive strength, which is related to the increase in calcium caused by the addition of CH [23]. CH provides active calcium for polymerization and promotes the formation of hydrated calcium silicate, thus improving the compressive strength of the specimen. However, with increasing CH content, the compressive strength of specimens LYY-12 and LYY-13 decreased gradually. This shows that excessive addition of CH is unfavorable for the development of compressive strength, and CH shows a strong water absorption ability. The addition of excessive CH leads to an increase in water absorption in the specimens. Therefore, a CH content of 20 g works the best.

### 3.3. Analysis of Dry and Wet Density and Water Absorption

#### 3.3.1. Cement Paste Specimens

Figure 4 shows the dry and wet densities and the water absorption of the specimens. The dry and wet densities of specimen LYY-1 are 1.76 and 1.92 g cm^−3^, respectively, and the water absorption reaches 9.19%. Although LYY-1 exhibits excellent mechanical properties, it does not meet the requirements of lightweight materials (dry density less than 1.6 g cm^−3^). When 10 g (1%) of foaming agent was added, the dry and wet densities of specimen LYY-2 decreased to 1.58 and 1.75 g cm^−3^, respectively, which are 10.2 and 8.9% lower than those of specimen LYY-1. However, the water absorption of specimen LYY-2 increased to 10.86% and 18.2% above specimen LYY-1. Although the addition of a foaming agent significantly decreased the dry and wet densities and thus met the requirements of lightweight materials, the introduction of a large number of bubbles led to a significant increase in water absorption. This, in turn, resulted in decreasing thermal insulation performance, mechanical properties, and the durability of the specimens.

Figure 4a demonstrates that when the content of the foaming agent increases to 20 g (2%), the dry and wet densities of specimen LYY-4 decrease to 1.04 and 1.25 g cm^−3^, respectively, which are 40.9 and 34.9% lower than those of specimen LYY-1, respectively. However, the water absorption of specimen LYY-4 increases to 20.29 and 120.8% above specimen LYY-1. With the increasing content of the foaming agent, the dry and wet densities of the specimen continue to decrease significantly; however, it also leads to a significant increase in water absorption. In summary, in terms of performance, cost, and environmental impact, the use of lightweight cement materials is neither economical nor able to achieve low-carbon requirements, energy-saving, and environmental protection characteristics.

#### 3.3.2. Bottom Ash Alkali-Activated Paste Specimen

Figure 4b illustrates that when the BA-alkali excitation specimen is used, the dry and wet densities of specimen LYY-5 decrease to 1.48 and 1.61 g cm^−3^, respectively, which are 15.9 and 16.1% lower than those of specimen LYY-1 (without a foaming agent) and 6.3 and 8% lower than those of specimen LYY-2 (mixed with 1% foaming agent). The BA alkali-activated specimen exhibits not only the advantage of better compressive strength but also better dry and wet densities than cement lightweight material. First, the BA is light in weight, and its density is significantly lower than that of cement (1.97 g cm^−3^ < 3.1 g cm^−3^), which has clear advantages of lower weight. More importantly, BA has self-foaming characteristics and expands in the case of alkali, which can significantly reduce the density of the specimen.

BA contains a small amount of metal aluminum; therefore, a redox reaction occurs in the case of strong alkali, releasing a large amount of hydrogen [22]. With the condensation of the specimen, numerous bubbles are sealed inside the specimen, resulting in volume expansion and thus low weight. This is the reason why the density of specimen LYY-5 is lower than that of specimen LYY-2. However, the water absorption of specimen LYY-5 reaches 10.9%, which is 18.3% higher than that of specimen LYY-1, but it is basically consistent with specimen LYY-2. This shows that the generation of bubbles inevitably leads to an increase in water absorption.

With increasing BA content, the dry and wet densities of specimens LYY-6 and LYY-7 continued to decrease, but the decreasing range gradually narrowed. The increase in BA content led to a continuous reduction in the quality and density of specimens and also led to a decrease in compressive strength. The increasing BA content significantly increased the foaming amount and led to a notable reduction in the density. In conclusion, the research and development of BA alkali-activated lightweight materials not only highlights the low-density advantages of BA but also achieves the lightweight effect without a foaming agent, thus turning waste into treasure and realizing the large-scale safe consumption and recycling of BA. This research is of great significance for China’s low-carbon environmental protection, energy conservation, and emission reduction strategy. However, the water absorption of specimens LYY-6 and LYY-7 increases strongly. The water absorption of specimen LYY-7 even exceeds that of specimen LYY-3. Therefore, water absorption must be controlled.

#### 3.3.3. Influence of Liquid Sodium Silicate

When the content of LSS reached 100 g, the dry and wet densities of specimen LYY-8 decreased to 1.21 and 1.35 g cm^−3^, which are 10.9 and 10.5% lower than those of specimen LYY-7 (without LSS), respectively. Moreover, when the content of LSS reached 200 g, the dry and wet densities of specimen LYY-9 further decreased to 1.18 and 1.31 g cm^−3^, which are 2.5 and 3.0% lower than those of specimen LYY-8, respectively. The addition of LSS significantly reduces the density (Figure 4c). Although BA can produce a large number of bubbles in the case of alkali, it cannot effectively seal the bubbles in the specimen. The bubbles can only be fixed when the specimen utilizes the initial setting. The initial setting time of the BA alkali-activated specimen is below 0.5 h, which leads to the overflow of bubbles at the early stage and cannot effectively cause the lightweight effect. LSS can significantly accelerate the condensation of alkali-activated materials (initial setting time 10 min) and reach the state of initial setting and final setting (both of which are reached at the same time), thus significantly improving the sealing efficiency of bubbles and further reducing the density of the specimen. This is attributed to the fact that the LSS contains a large amount of active silicon, which increases the polymerization rate, accelerates the polymerization process, promotes the formation of calcium silicate hydrate and calcium aluminosilicate gel, and finally accelerates condensation [24].

At the same time, the water absorption of specimen LYY-8 decreased to 12.3%, which is 25.3% lower than that of specimen LYY-7 (without LSS), indicating that LSS also has a water absorption reducing function. As LSS is a viscous liquid with good sealing and isolation function, it can seal and isolate microbubbles into independent bubble units, block penetration and linkage between bubbles, effectively prevent the infiltration of free water, and reduce overall water absorption [25]. Therefore, LSS not only plays an effective role in bubble fixation but also blocks the penetration between bubbles, which further reduces density and water absorption.

With increasing LSS content, the performance of specimen LYY-10 did not improve compared to specimen LYY-9; however, water absorption still decreased significantly. Continuous increases in the LSS content could continue to reduce both density and water absorption. The increase in LSS content can provide more active silicon, accelerate the reaction rate, accelerate the formation of polymerization products, improve the sealing bubble efficiency, and reduce the density. However, there is a certain limit to the effect that LSS can exert on promoting coagulation. If the LSS content exceeds 25% of the cementitious material, coagulation is not further accelerated [26]. Therefore, the dry and wet densities of specimen LYY-10 (LSS content 31%) no longer continue to decrease, but the sealing and blocking effect of LSS on bubbles is further strengthened, thus strengthening the sealing effect. This is the reason why water absorption continues to decline.

#### 3.3.4. Effect of Calcium Hydroxide

The dry and wet densities of specimen LYY-11 (CH mixed with 20 g) decreased to 1.15 and 1.25 g cm^−3^, which are 4.2 and 3.8% lower than those of specimen LYY-9 (without CH). The addition of CH continues to reduce the density (Figure 4d). CH can provide active calcium for the polymerization environment via dissolution; therefore, active calcium and LSS favor the active repolymerization of reactive silicon, thus forming hydrated calcium silicate gel, accelerating the condensation of specimens, enhancing the bubble sealing ability, and achieving a further reduction in density [27]. However, CH also causes the water absorption of specimen LYY-11 to rise to 12.1%, which is slightly higher than that of specimen LYY-9. This is caused by the good water absorption of CH. With the increasing CH content, the dry and wet densities of specimens LYY-12 and LYY-13 continued to increase, and the water absorption increased significantly. Therefore, the excessive addition of CH could not continuously reduce the dry and wet densities but led to a significant increase in water absorption.

### 3.4. Optical Microscopy Analysis

#### 3.4.1. Cement Lightweight Specimen

Specimen LYY-1 (without a foaming agent, Figure 5a) exhibits a relatively flat microstructure without apparent bubbles or cracks on the surface, and the bonding between particles is relatively dense. With the addition of a foaming agent, the microstructure of specimen LYY-2 (Figure 5b) shows not only bubbles and cracks of different sizes and shapes but also apparent defects. The pore diameter of the bubbles is between 100 and 500 µm. At the same time, the microparticles are not closely bonded, and the overall structure is uneven. With the increasing content of the foaming agent, the microstructure defects of specimen LYY-4 (Figure 5c) become more severe, and the pore diameter of the bubbles increases to 500 and 1000 µm; even millimeter-scale defects appear. This shows that with the increase in foaming agents, the internal defects in the specimens increase significantly.

Owing to the large difference between the densities of foaming agent and cement paste, the addition of foaming agent cannot ensure that the generated bubbles are evenly distributed throughout the specimen, resulting in the formation of pores and cracks. With the continuous merging of bubbles, penetrating pores and cracks are formed, which results in a large number of defects, a sudden drop in compressive strength, and a significant increase in water absorption capability. More seriously, the uneven distribution of bubbles leads to the discreteness of specimens in many aspects, such as compressive strength, unit volume density, water absorption, thermal insulation, and durability. These aspects ultimately result in great differences in the performance of each part of the same material, which severely affects the use of the material. Therefore, lightweight cement materials do not show advantages in terms of preparation cost, material performance, energy conservation, and environmental protection.

#### 3.4.2. Bottom Ash Alkali-Activated Light Specimen

When BA is used to prepare lightweight materials, uniform bubbles and pores appear, as shown in the microstructure of specimen LYY-5 (Figure 5d). The microstructure of specimen LYY-5 is flat, the bonding between particles is relatively tight, and there are no apparent defects. Moreover, the pore size of bubbles is mainly concentrated between 100 and 600 µm, and there are also a small number of millimeter-sized bubbles with a relatively uniform distribution. Therefore, the interior of the lightweight specimen prepared with BA foamed and expanded evenly, which reduced the impact on the compressive strength, ensured the homogeneity, and provided better thermal insulation and durability. As BA was pretreated by crushing and ball milling, metal aluminum could be evenly distributed in BA, and all parts of the specimen were evenly foamed, resulting in overall expansion [28]. This is the main reason why bubbles are evenly distributed and why the compressive strength of BA alkali-activated lightweight specimens is better than that of cement lightweight specimens.

With an increasing BA content, the bubbles in the specimens LYY-6 and LYY-7 increased significantly (as shown in Figure 5e,f); however, they were still evenly distributed without penetrating cracks and pores and serious defects. The diameter of the bubbles also did not increase. Increasing the content of BA could significantly increase the amount of foaming, which is responsible for the continuous decrease in dry and wet densities. The significant increase in bubble content is bound to reduce compressive strength, even though the bubbles can still be evenly distributed, which tremendously reduces the impact on compressive strength.

#### 3.4.3. Influence of LSS and CH

With the addition of LSS and CH, the microbubbles in specimens LYY-9 and LYY-11 increased, and a small number of micro-cracks appeared, while these defects did not appear in specimens LYY-5, LYY-6, and LYY-7. The specimens expanded clearly as a whole, in the shape of a sponge (as shown in Figure 5g,h). The bubble volume was significantly reduced, and the exposed pore opening was also reduced (compared with specimens without LSS). The diameter of most of the pores was also reduced to between 100 and 300 µm. First, LSS and CH could accelerate the condensation, improve the sealing efficiency of bubbles, reduce the overflow of bubbles, and promote the further expansion of the volume, thus reducing the dry and wet densities [29]. More importantly, LSS is a viscous liquid with a retraction function. When bubbles emerged from LSS, the size of the remaining pores was smaller, which helped to reduce water absorption [30]. After condensation and solidification, the LSS could cover the surface of microparticles and play an effective sealing and waterproofing role. This is the reason why the water absorption of the specimen decreased with the increasing LSS content.

### 3.5. Scanning Electron Microscopy Analysis

At a magnification of 100 times, the distribution of bubbles and pores in specimen LYY-11 was relatively uniform (Figure 6a), the overall structure was also relatively uniform, and the bonding between the microparticles was relatively loose. This shows that the foaming expansion of BA in the case of alkali could ensure that the bubbles were evenly distributed in the specimen, avoiding the uneven distribution of bubbles caused by the foaming agent, thus reducing the internal defects and ensuring better mechanical properties and durability [31].

At magnifications of 500 and 1000 times (Figure 6b,c, respectively), the pore diameter in specimen LYY-11 was small (both between 20–200 μm, which is significantly smaller than the foaming diameter of the foaming agent), which existed independently. Notably, the phenomenon of penetration into the pore group or pore aggregation and the formation of local defects was not observed, which is related to the sealing effect of LSS and CH. The existing microbubbles not only reduce the dry and wet densities of the specimen but also minimize the negative effects on the mechanical properties, thermal insulation, and durability and control the improvement of water absorption. Furthermore, the particles around the micropores become wrapped with LSS paste to form a uniform gel, which further reduces the stomata [32]. This indicates that LSS plays a role in sealing the stomata, thereby reducing water absorption.

At a magnification of 5000 times (Figure 6d), a large number of nanopores and pores between microparticles can be observed. This shows that the bubbles produced by BA are mainly micro- or nano-sized, while few pores are above the millimeter level, and few large defects are produced. Nanopores do not adversely affect the mechanical properties and water absorption capacity of the specimen [33]. Therefore, regardless of the product preparation cost, energy conservation, environmental protection, and material performance, BA alkali-activated lightweight specimens outperform cement-foamed specimens.

## 4. Conclusions

In this study, self-foaming alkali-activated lightweight material was prepared by the pretreatment of municipal solid waste incineration bottom ash (BA). Based on the results of this study, the following conclusions can be drawn:(1)Owing to the uneven distribution of bubbles, lightweight cement materials prepared using a foaming agent produced penetrating pores and cracks. These adversely impacted the compressive strength, thermal insulation, and the durability of the specimens. With the increasing content of the foaming agent, the defects became increasingly severe, ultimately resulting in the sudden decline of all aspects of the properties.(2)BA exhibits the characteristics of lightweight and alkali-foaming expansion, and the generated bubbles were evenly distributed, thus avoiding the formation of defects and alleviating the impact of the foaming expansion on all aspects of the properties. With an increasing BA content, the foaming effect was increased significantly, and the dry and wet densities of the specimen continued to decrease. However, water absorption increased significantly, but the declining range of compressive strength was significantly lower than that of cement foaming specimens.(3)LSS could provide active silicon for the polymerization reaction, promote the reaction progress, and accelerate the formation of polymerization products. This accelerated the condensation of specimens, improved the bubble sealing efficiency, reduced the opening diameter of bubbles, further reduced the dry and wet densities and water absorption of specimens, and alleviated the decline of compressive strength. With an increasing LSS content, the dry and wet densities of the specimen first decreased and then basically remained unchanged; however, the water absorption gradually decreased further.(4)CH provides active calcium for polymerization, promotes the formation of hydrated calcium silicate gels, further accelerates the condensation of specimens, enhances the efficiency of foam fixation, and further reduces the dry and wet densities. However, CH readily absorbs water, which leads to an apparent increase in water absorption. With an increasing LSS content, the dry and wet densities increased slowly; however, the water absorption exhibited a clearly increasing trend.

## Figures and Tables

**Figure 1 polymers-14-02222-f001:**
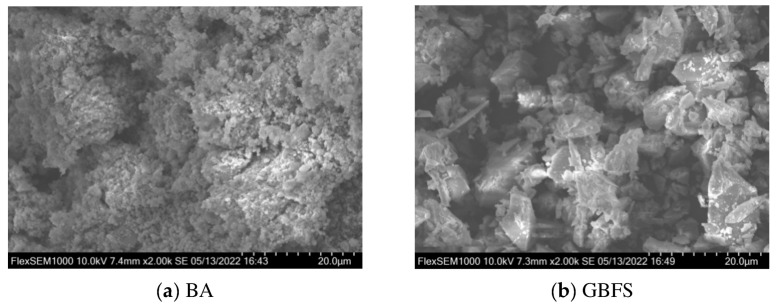
Micromorphology of BA and GBFS (2000×).

**Figure 2 polymers-14-02222-f002:**
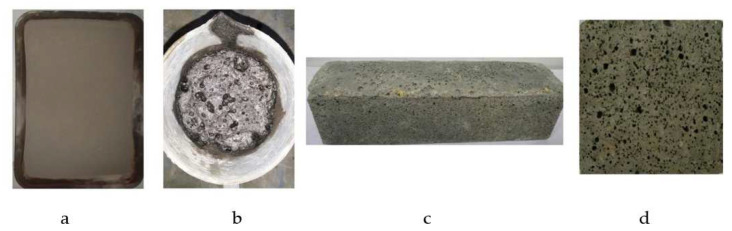
Examples of light materials prepared using BA. (**a**) BA after pretreatment, (**b**) Foams and expansion phenomenon, (**c**) Appearance of specimen LYY-9, and (**d**) Surface pore morphology of specimen LYY-9.

**Figure 3 polymers-14-02222-f003:**
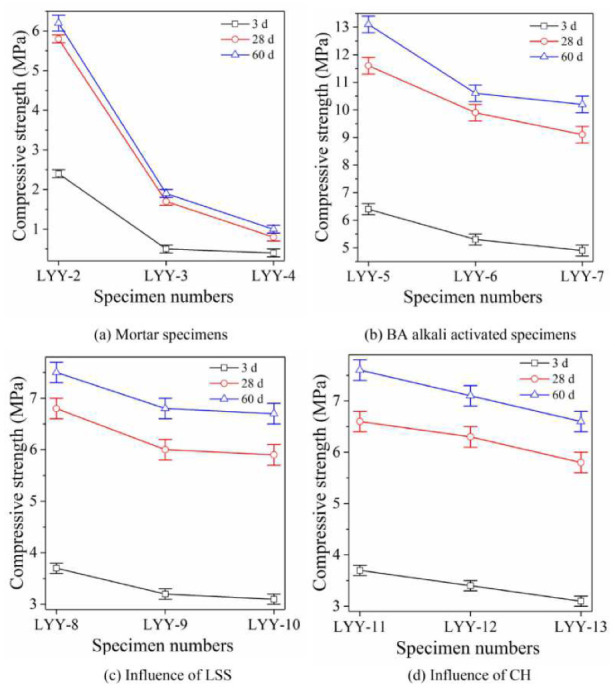
Analysis of mechanical properties of cement and bottom ash alkali-activated materials.

**Figure 4 polymers-14-02222-f004:**
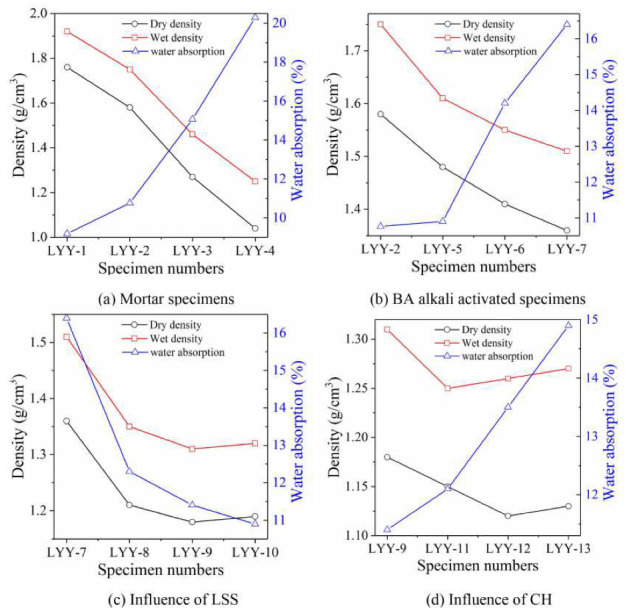
Analysis of dry and wet densities and water absorption.

**Figure 5 polymers-14-02222-f005:**
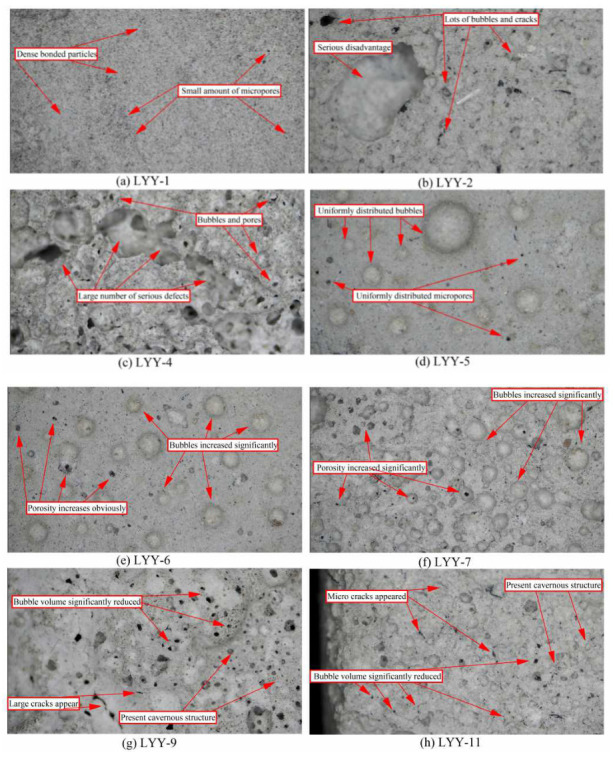
Microstructure of specimens (200×).

**Figure 6 polymers-14-02222-f006:**
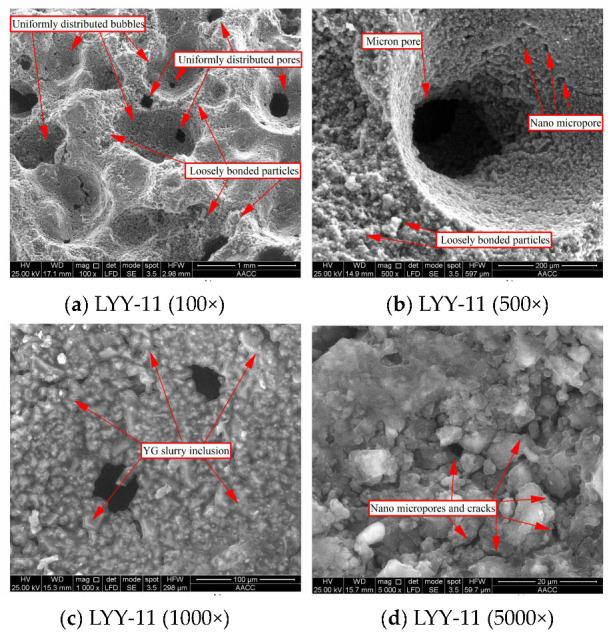
Scanning electron microscopy analysis of BA alkali-activated specimen.

**Table 1 polymers-14-02222-t001:** XRF spectroscopy detection of raw materials (%).

Raw Materials	SiO_2_	CaO	Al_2_O_3_	Fe_2_O_3_	MgO	K_2_O	Na_2_O	Others	Loss
BA	53.8	14.4	14.2	6.2	3.3	2.5	2.2	0.6	1.6
GBFS	31.4	34.7	18.7	0.6	9.3	-	-	1.8	0.7

**Table 2 polymers-14-02222-t002:** Mix proportion/g.

Specimen Numbers	Cement	BA	GBFS	CH	LSS	NH	Foaming Agent	Water	Liquid–Solid Ratio
LYY-1	1000	0	0	0	0	0	0	500	0.5
LYY-2	1000	0	0	0	0	0	10	500	0.5
LYY-3	1000	0	0	0	0	0	20	500	0.5
LYY-4	1000	0	0	0	0	0	30	500	0.5
LYY-5	0	400	600	0	0	40	0	500	0.5
LYY-6	0	500	500	0	0	40	0	500	0.5
LYY-7	0	600	400	0	0	40	0	500	0.5
LYY-8	0	600	400	0	100	40	0	435	0.5
LYY-9	0	600	400	0	200	40	0	370	0.5
LYY-10	0	600	400	0	300	40	0	305	0.5
LYY-11	0	600	380	20	200	40	0	370	0.5
LYY-12	0	600	350	50	200	40	0	370	0.5
LYY-13	0	600	300	100	200	40	0	370	0.5

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
