# Peer review of "Preparation and Properties of Municipal Solid Waste Incineration Alkali-Activated Lightweight Materials through Spontaneous Bubbles"

_polymers, 2022, doi:10.3390/polym14112222_

Round 1
Reviewer 1 Report
The research paper deals with the municipal solid waste incineration alkali activated lightweight materials through spontaneous bubbles.
Using the municipal solid waste incineration bottom ash is a very actual and interesting theme. I suggest to authors to make the paper more suitable for Polymers journal or try to publish it in Materials journal.
Among other things, I suggest the following corrections of the paper:
1. Introduction - You should emphasize why you would like to publish the paper in "Polymers" - you should describe use of polymers in alkali activated lightweight materials in more detail. Otherwise, the paper should be more suitable for Materials journal.
-
2. Experimental materials and Methods
- line 81 - you should specify the type of ball mill
- line 84 - I think you meant silicone oxide,not "silicone"
- Table 1 - What type of los? Is it Loss on ignition?
- SEM images of DH and KZ should be added, regarding DH I also suggest to state content of pollutants (heavy metals, etc.)
- you should use past tense
- You should specify used Portland cement in more detail
- Used foaming agent must be specified
- Description of he determination of dry and wet density and water absorption is unnecessarily long
- You should also test porosity of the samples...
3. Results and Discussion
- Figure 2 - space is missing between the unit and "Compressive strength", same in Figure 3
- Optical microscopic analysis - you should state pores diameter in all images and describe causes of disadvantages and defects in more detail...
- Figure 5 - (1000x) instead of (x1000) and the scale is difficult to see
line 451- You mention formation of polymerization products, but what type of polymerization products???
Author Response
Response to the Reviewers
The authors would like to thank all the reviewers for their great comments on the manuscript. Their comments have been taken into consideration seriously while revising the manuscript. The revisions are underlined in the revised manuscript for easier tracking. Their comments/concerns are addressed as follows. Note that the line numbers referred to by the reviewers may change due to the revision of the manuscript.
Reviewers' comments:
Reviewer #1:
- Introduction - You should emphasize why you would like to publish the paper in "Polymers" -you should describe use of polymers in alkali activated lightweight materials in more detail. Otherwise, the paper should be more suitable for Materials journal.
It has been revised according to the reviewer's suggestions. Page 2, line 43-55.
- Experimental materials and Methods
- line 81 - you should specify the type of ball mill.
It has been revised according to the reviewer's suggestions. Page 3, line 72-73.
- line 84 - I think you meant silicone oxide,not "silicone"
It has been revised according to the reviewer's suggestions. Page 3, line 75.
- Table 1 - What type of los? Is it Loss on ignition?
Yes,it is.
- SEM images of DH and KZ should be added, regarding DH I also suggest to state content of pollutants (heavy metals, etc.)
It has been revised according to the reviewer's suggestions. Page 4, line 100.
Because the bottom ash is specified as recyclable waste in China, and this paper mainly analyzes the performance of bottom ash light materials, not the environmental pollution characteristics of bottom ash, there is no analysis on the content and precipitation characteristics of heavy metals in bottom ash, but the author will focus on this problem in the next paper.
- you should use past tense
The language has been polished by a special translation agency. Of course, further amendments have been made to the questions raised by the reviewers.
- You should specify used Portland cement in more detail
It has been revised according to the reviewer's suggestions. Page 3, line 94-95.
- Used foaming agent must be specified
It has been revised according to the reviewer's suggestions. Page 3, line 96-100.
- Description of he determination of dry and wet density and water absorption is unnecessarily long
Dry and wet density and water absorption are important factors affecting the properties of lightweight materials, and the measurement methods are directly related to the differences of the above properties. At the same time, the measurement specifications in different countries are different. Therefore, in order to let readers accurately understand how the data is obtained, the measurement methods of dry and wet density and water absorption are described in detail here.
- You should also test porosity of the samples...
This article mainly describes the preparation of lightweight materials without adding foaming agent by making full use of the foaming characteristics of DH in alkali, and compares them with the lightweight materials prepared with foaming agent. The range of pore diameter measured by mercury injection method and nitrogen adsorption method is 1 nm-500 µm. However, the pore diameter of DH foaming in case of alkali is between 1mm-5mm, and the bubbles will continue to gather and grow inside the sample, which obviously exceeds the measurement range of pore diameter. Therefore, the porosity is not analyzed in this paper
- Results and Discussion
- Figure 2 - space is missing between the unit and "Compressive strength", same in Figure 3
It has been revised according to the reviewer's suggestions. Page 8, line 240, Page 11, line 305.
- Optical microscopic analysis - you should state pores diameter in all images and describe causes of disadvantages and defects in more detail...
Since the optical microscope does not have the function of directly measuring the pore size, the pore size can only be estimated according to the magnification. Moreover, the author strengthened the analysis of pore diameter and the discussion of pore harm. Page 12-14, line 359-360,362-365,381-382,392,405-406.
- Figure 5 - (1000x) instead of (x1000) and the scale is difficult to see
line 451
It has been revised according to the reviewer's suggestions. Page 15, line 420. The scale is displayed in the lower right corner of the diagram.
- You mention formation of polymerization products, but what type of polymerization products???
Polymerization products refer to hydrated calcium silicate and hydrated calcium aluminosilicate gel. Under strong alkaline excitation, active calcium, silicon aluminum and other substances in KZ and DH are continuously dissolved and polymerized to form hydrated calcium silicate and hydrated calcium aluminosilicate gel.

Reviewer 2 Report
The manuscript deals with an important research topic. The pretreatment of municipal solid waste incinerator bottom ash (MSWIBA) has been used to produce self-foaming alkali-activated lightweight materials. The manuscript needs essential modifications before it is accepted for publication, as follows:
- The English language and the structure of the manuscript must be improved.
- All abbreviations used in the text must be modified to be common. For example, the following abbreviations must be changed: DH, YG, and KZ.
- Some relevant references in this area are still missing in the Introduction section, so at least five important references from recent years must be added, see for example:
- https://doi.org/10.3390/app1207353
- https://doi.org/10.1016/j.conbuildmat.2021.123814
- Sections 2 (Experimental Work) and 3 must be revised and rearranged.
- Section 2.2.1. What is the scientific basis on which the ingredients and their proportions were selected? This must be explained in detail
- What are the specifications based on which the dimensions of the specimens and the test procedures were adopted? They must be stated.
- Figures 2 and 3 must be improved.
Author Response
Response to the Reviewers
The authors would like to thank all the reviewers for their great comments on the manuscript. Their comments have been taken into consideration seriously while revising the manuscript. The revisions are underlined in the revised manuscript for easier tracking. Their comments/concerns are addressed as follows. Note that the line numbers referred to by the reviewers may change due to the revision of the manuscript.
Reviewers' comments:
Reviewer #2:
The manuscript deals with an important research topic. The pretreatment of municipal solid waste incinerator bottom ash (MSWIBA) has been used to produce self-foaming alkali-activated lightweight materials. The manuscript needs essential modifications before it is accepted for publication, as follows:
- The English language and the structure of the manuscript must be improved.
The language has been polished by a special translation agency.
- All abbreviations used in the text must be modified to be common. For example, the following abbreviations must be changed: DH, YG, and KZ.
It has been revised according to the reviewer's suggestions.
- Some relevant references in this area are still missing in the Introduction section, so at least five important references from recent years must be added, see for example:
- https://doi.org/10.3390/app1207353
- https://doi.org/10.1016/j.conbuildmat.2021.123814
It has been revised according to the reviewer's suggestions. Page 2, line 43-55.
- Sections 2 (Experimental Work) and 3 must be revised and rearranged.
- Section 2.2.1. What is the scientific basis on which the ingredients and their proportions were selected? This must be explained in detail
Since the size of the sample is 40 mm × 40 mm × 160 mm, when using 1000g cementitious material and 500 g water, a group of triple molds can be completely poured. Furthermore, the development of the mechanical properties mainly depends on the polymerization reaction of GBFS, and the foaming amount changes the content of BA. Therefore, by changing the content of GBFS and BA, it can not only have certain mechanical properties, but also achieve good lightweight effect.
- What are the specifications based on which the dimensions of the specimens and the test procedures were adopted? They must be stated.
It has been revised according to the reviewer's suggestions. Page 4, line113-114.
- Figures 2 and 3 must be improved..
It has been revised according to the reviewer's suggestions. Page 8, line 240, Page 11, line 305.

Round 2
Reviewer 1 Report
Dear authors,
the paper has been corrected according my comments, well done.
Reviewer 2 Report
The authors have successfully addressed all my comments. Therefore, I recommend the publication of this manuscript.